# ONE EXPLANATION DOES NOT FIT XIL

**Felix Friedrich**[1,2]**, David Steinmann**[1]**, Kristian Kersting**[1,2,3,4]
[1]Department of Computer Science, TU Darmstadt; [2]Hessian.AI;
[3]Centre for Cognitive Science, TU Darmstadt; [4]German Center for Artificial Intelligence (DFKI)
{lastname}@cs.tu-darmstadt.de

## ABSTRACT

Current machine learning models produce outstanding results in many areas but, at the same time, suffer from shortcut learning and spurious correlations. To address such flaws, the explanatory interactive machine learning (XIL) framework has been proposed to revise a model by employing user feedback on a model's explanation. This work sheds light on the explanations used within this framework. In particular, we investigate simultaneous model revision through multiple explanation methods. To this end, we identified that *one explanation does not fit XIL* and propose considering multiple ones when revising models via XIL.

**Motivation**   Nowadays, machine learning models generally suffer from flaws, e.g. model bias (Friedrich et al., 2023; Bender et al., 2021) or confounding behavior (Geirhos et al., 2020; Lapuschkin et al., 2019). Therefore, it becomes crucial to make models understandable as their applications get more and more integrated into our lives. As a remedy, explainable artificial intelligence (XAI) has emerged with methods to explain a model, often its outputs, to the user. One step further, several works leverage such explanations in the learning setting, to improve a model beyond explainability or unconfound it (Teso & Kersting, 2019; Teso et al., 2022; Selvaraju et al., 2019; Friedrich et al., 2022). Therein, user interaction plays a central role, substantially enhancing recent applications (Ouyang et al., 2022). A promising framework that leverages explanations interactively to improve a model's performance is XIL (*cf.* Fig. 1). Intuitively, given a model which provides an explanation (EXPLAIN, Fig. 1) for a decision of a selected example, the user can interact with the model and provide corrective feedback on the explanation.

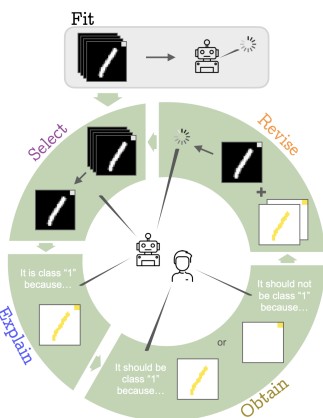

This way, the model is not only optimized for the actual task but also revised to align its explanations with the user-provided feedback. More precisely, a model is trained on a specific task (FIT) and (potentially) learns spurious correlations. XIL encourages the user to interact with the model, wherefore the user selects a suspicious training sample (SELECT). Next, the model provides a decision plus explanation for this sample (EXPLAIN). The user gives corrective feedback for the sample (OBTAIN). Finally, the model explanation is aligned with the user feedback to revise incorrect model behavior (REVISE). However, XIL's EXPLAIN module has only been realized and investigated for one explainer at a time (e.g., RRR (Ross et al., 2017) uses Input Gradients (IG)). This work transfers the *one explanation does not fit all* paradigm (Arya et al., 2019; Sokol & Flach, 2020) to XIL. In general, each explainer has inherent limitations, e.g., IG (Hechtlinger, 2016) provides only local explanations. In turn, revising a model via

Figure 1: XIL typology by Friedrich et al. (2022).

XIL implemented with a single explainer does not ensure a model revision in its entirety. That means explainers' different capabilities and limitations translate to XIL methods and impact their effectiveness (Friedrich et al., 2022). Therefore, we next investigate XIL with explainer combinations. We show this helps further improve model revision regarding explanation quality. This work motivates designing future methods that leverage explanations of multiple explainers, as no best explainer exists to be optimized for.

**Methods**   Previous approaches (Ross et al., 2017; Schramowski et al., 2020; Shao et al., 2021) were already leveraging explanations to revise a model. They usually follow the paradigm of optimizing two objectives at the same time: the prediction ($\mathcal{L}^{pred}$) and explanation loss ($\mathcal{L}^{xil}$). The

| a) Acc (↑) | train | test |
|---|---|---|
| w/o decoy | $99.8_{\pm0.1}$ | $98.8_{\pm0.1}$ |
| Vanilla | $99.9_{\pm0.0}$ | $78.9_{\pm1.1}$ |
| RRR | $99.9_{\pm0.1}$ | $98.8_{\pm0.1}$ |
| RRR-G | $99.7_{\pm0.2}$ | $97.4_{\pm0.7}$ |
| RBR | $\mathbf{100.0}_{\pm0.0}$ | $\mathbf{99.1}_{\pm0.1}$ |

| b) WR (↓) | Vanilla | RRR | RRR-G | RBR |
|---|---|---|---|---|
| IG | $23.1_{\pm3.8}$ | $\mathbf{0.0}_{\pm0.0}$ | $11.9_{\pm2.1}$ | $2.0_{\pm1.3}$ |
| GradCAM | $38.7_{\pm4.6}$ | $13.3_{\pm2.0}$ | $\mathbf{1.5}_{\pm0.8}$ | $15.2_{\pm3.8}$ |
| IntGrad | $41.3_{\pm2.2}$ | $\mathbf{0.0}_{\pm0.0}$ | $21.0_{\pm3.4}$ | $17.8_{\pm5.4}$ |
| LIME | $59.8_{\pm2.0}$ | $\mathbf{32.1}_{\pm0.4}$ | $33.3_{\pm2.8}$ | $37.7_{\pm3.0}$ |

Table 1: Mean accuracy and WR scores [%] with standard deviation (5 runs) on DecoyMNIST. All XIL methods overcome the confounder in terms of accuracy (a). However, the WR scores (b) show that XIL mainly improves the explanation quality for internally-used explainers. Best values bold.

former is the same as in the standard training objective, while the explanation loss additionally constrains the explanations based on user feedback. The combined loss enforces a concurrent optimization of model outputs and explanations. So far, $\mathcal{L}^{\text{xil}}$ was only implemented with a single explainer (EXPLAIN). For example, RRR (Ross et al., 2017) uses IG (Hechtlinger, 2016), RBR (Shao et al., 2021) uses influence functions (IF, Koh & Liang (2017)), while RRR-G (Schramowski et al., 2020) use gradient-weighted class activation maps (GradCAM, Selvaraju et al. (2017)). In contrast to these methods, we realize $\mathcal{L}^{\text{xil}}$ with multiple explainers, giving

$$\mathcal{L} = \mathcal{L}^{\text{pred}} + \sum_i \lambda_i \mathcal{L}_i^{\text{xil}} \tag{1}$$

**Results**  We base our experimental evaluation on the DecoyMNIST dataset —a variation of MNIST with decoy squares in the image corners, confounding the training data. We measure model performance with prediction accuracy and the explanation quality via a wrong reason measure (WR, *cf.* A.1; lower is better). It examines how wrong a model's explanation for a specific prediction is, given ground-truth wrong reasons. Further details and results can be found in A.2 and A.3.

Tab. 1a shows that XIL methods independent of the internally-used explainer successfully revise a model in terms of accuracy. However, Tab. 1b demonstrates that the model still relies on wrong reasons when generating explanations with various explainers. For example, applying RRR (employing IG explanations) substantially reduces WR for IG and Integrated Gradients (IntGrad), but GradCAM and LIME (Ribeiro et al., 2016) scores are still high, i.e. have high activations in the confounder area. This highlights that the WR score of the internally-used explainer alone is no suitable indicator for confounding behavior. More importantly, the results show that revising a model with XIL through one explainer does not generalize to (all) different explainers. In contrast, Tab. 2 illustrates that leveraging a combination of various explanations into one XIL method reduces WR among multiple explainers while the accuracy remains on par (Tab. 3). Combining RRR and RRR-G yields low WR scores for all explainers (except LIME, though improved), where single methods struggle with. Moreover, combining RRR and RBR shows that not directly related explainers (GradCAM or LIME) can be improved, too. The final combination again highlights that combining multiple explanations better *fits* XIL, i.e., further improving a model's explanation quality. However, one can see that combining methods does not set all scores to zero. This questions the reliability and robustness of explainers, an active research area (Adebayo et al., 2018). Hence, more research on explainers is needed. Furthermore, as the rightmost column in Tab. 2 shows (GradCAM score is not lowest), another exciting avenue for future work entails further investigating $\lambda_i$ to trade off the influence of each explainer. Finally, the increase in computational cost must be kept in mind.

| | RRR+RRR-G | RRR+RBR | RRR-G+RBR | RRR+RRR-G+RBR |
|---|---|---|---|---|
| IG | $\mathbf{0.0}_{\pm0.0}$ | $\mathbf{0.0}_{\pm0.0}$ | $1.0_{\pm0.1}$ | $\mathbf{0.0}_{\pm0.0}$ |
| GradCAM | $3.1_{\pm1.7}$ | $11.8_{\pm2.9}$ | $\mathbf{2.3}_{\pm1.5}$ | $3.5_{\pm2.5}$ |
| IntGrad | $2.2_{\pm0.1}$ | $\mathbf{0.0}_{\pm0.0}$ | $13.5_{\pm0.1}$ | $\mathbf{0.0}_{\pm0.0}$ |
| LIME | $29.6_{\pm0.8}$ | $31.0_{\pm0.9}$ | $33.1_{\pm0.8}$ | $\mathbf{27.9}_{\pm1.0}$ |

Table 2: Mean WR scores [%] with sd (5 runs) on DecoyMNIST. The columns depict combinations of explainers used for XIL. The WR scores of combined methods are lower than for methods based on single explainers (*cf.* Tab. 1b). Lower is better; best values bold.

**Conclusion**  In this work, we studied XIL's performance from the perspective of explanation methods. We found that optimizing for a single explanation method *does not fit* XIL. Instead, combining different explanation methods through simultaneous optimization further improves explanation quality, even beyond the optimized explanation methods. Emphasizing the complexity of faithful and explainable models, our results contribute to this goal and motivate future research.

URM STATEMENT

The authors acknowledge that at least one key author of this work meets the URM criteria of ICLR 2023 Tiny Papers Track. Authors FF and DS meet the URM criteria of ICLR 2023 Tiny Papers Track.

ACKNOWLEDGEMENTS

The authors thank Raynard Widjaja for the preliminary results. This work benefited from the Hessian Ministry of Science and the Arts (HMWK) projects "The Third Wave of Artificial Intelligence - 3AI", "The Adaptive Mind" and Hessian.AI, the "ML2MT" project from the Volkswagen Stiftung as well as from the ICT-48 Network of AI Research Excellence Center "TAILOR" (EU Horizon 2020, GA No 952215).

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

# A  APPENDIX

Our code is publicly available at `https://github.com/ml-research/A-Typology-for-Exploring-the-Mitigation-of-Shortcut-Behavior/tree/extended`

## A.1  WR MEASURE

Models revised with XIL are often evaluated based on their accuracy. However, assessing the accuracy does not cover explanation quality, and is not sufficient to capture the effect of XIL. Therefore, we also use the WR measure proposed in Friedrich et al. (2022). For an input $X$, the WR score measures the percentage of activated pixels from a model explanation ($expl(X)$) compared to a user annotation mask $M$. In this work, the metric compares how much the model explanation relies on wrong reasons, i.e. activates pixels in the confounder area which represents features of shortcut learning/ spurious correlation.

$$\text{WR}(X, M) = \frac{sum(b_\alpha(norm^+(expl(X))) \circ M)}{sum(M)}, \tag{2}$$

In this equation, $\circ$ is the Hadamard product. $norm^+$ normalizes the explanation of explainer $expl$ to $[0, 1]$, while only accounting for positive values. The normalized explanation is then binarized by $b_\alpha$, with $\alpha$ as threshold ($expl_{ij} > \alpha \Rightarrow 1$ else $0$).

## A.2  EXPERIMENTAL PROTOCOL

Our focus is on evaluating computer vision datasets, which are an active area of research due to their well-known confounders (Zhong & Ettinger, 2017). In these datasets, confounders are visual regions in images (e.g. colored corners) that correlate with image class but are not causal factors for determining the true class. Confounders can deceive the model and lead to shortcut learning rules. In our standard setup, we train machine learning models on a confounded train set and test them on a non-confounded test set with the goal of guiding the model to ignore the confounder. To assess different facets of XIL, we chose two benchmark datasets: Decoy(F)MNIST. These datasets have visually separable confounders that provide a controlled environment for evaluation.

Our experiments use a CNN with two convolution layers (channels=[20,50], kernel size=5, stride=1, pad=0), each followed by a ReLU activation and max-pooling layer. The last two layers are fully-connected. We optimize our models using Adam and a learning rate of 0.001 for 50 epochs. For the standard experiments, we apply the explanation loss from the beginning. We use the standard train-test split (60000 to 10000) and set the batch size to 256.

The DecoyMNIST dataset (Ross et al., 2017) is a modified version of the MNIST dataset that introduces decoy squares into the train set. These $4 \times 4$ gray squares appear in randomly chosen corners and their shades are functions of their digits. The gray-scale corners and colors are randomized in the test set. Binary feedback masks $M$ mark confounders for the penalty strategy, while the masks mark the digits (relevant region) for the reward strategy. FashionMNIST (FMNIST) (Xiao et al., 2017) is an updated version of MNIST that is more complex and less overused in research. FM-NIST contains images from ten different fashion article classes. The DecoyFMNIST dataset also introduces confounding squares in the same manner as DecoyMNIST.

## A.3  FURTHER EXPERIMENTAL RESULTS

Here, we present further results. We evaluated our method on the DecoyMNIST and DecoyFMNIST datasets. We based our experimental evaluation on the work of Friedrich et al. (2022) and more details can be found there. We used the XIL methods RRR, RRR-G, RBR, CDEP (Rieger et al., 2020), HINT (Selvaraju et al., 2019) and counterexamples (CE (Teso & Kersting, 2019)). Notably, HINT does utilize a different feedback revision strategy (REVISE, *cf.* Fig. 1) than the other methods, as it encourages the model to attend to the regions of the user feedback. CE utilizes dataset augmentation and does not apply a different loss to the model, and instead adds corrected examples to the training dataset.

| XIL | DecoyMNIST | | DecoyFMNIST | |
|---|---|---|---|---|
| | train | test | train | test |
| Vanilla | 99.9 | 78.9 | 99.5 | 58.3 |
| w/o decoy | 99.8 | 98.8 | 98.7 | 89.1 |
| RRR | 99.9 | 98.8 | 98.7 | **89.4** |
| RRR-G | 99.7 | 97.4 | 90.2 | 78.6 |
| RBR | 100.0 | **99.1** | 96.6 | 87.6 |
| CDEP | 99.3 | 97.1 | 89.8 | 76.7 |
| HINT | 97.6 | 96.6 | 99.0 | 58.2 |
| CE | 99.9 | 98.9 | 99.1 | 87.7 |
| RRR + RBR + RRR-G | 98.8 | 98.3 | 92.0 | 88.4 |
| RRR + RRR-G | 99.2 | 98.1 | 91.7 | 86.9 |
| RRR + RBR | 99.9 | **98.9** | 98.5 | **89.5** |
| RRR + CDEP | 99.3 | 98.5 | 98.2 | 88.4 |
| RBR + CDEP | 99.2 | 98.6 | 87.0 | 87.0 |
| RRR-G + CDEP | 98.0 | 96.6 | 86.3 | 81.2 |
| RRR-G + HINT | 93.2 | 94.6 | 89.6 | 77.3 |
| CDEP + HINT | 97.7 | **97.4** | 89.8 | 75.9 |
| RRR + HINT | 93.4 | 94.4 | 93.4 | **89.4** |
| RBR + HINT | 95.0 | 95.4 | 94.5 | 84.9 |
| RRR + CE | 99.9 | 98.8 | 99.2 | **89.3** |
| RBR + CE | 100.0 | **99.1** | 94.0 | 85.2 |
| RRR-G + CE | 97.7 | 96.7 | 94.6 | 86.4 |
| CDEP + CE | 99.7 | 98.4 | 94.1 | 86.1 |
| HINT + CE | 98.4 | 97.4 | 93.0 | 87.3 |

Table 3: Mean accuracy scores [%] of XIL methods and combinations (5 runs) for Decoy(F)MNIST. The first part shows the accuracy of the confounded Vanilla model, a model on non-confounded data followed by single XIL methods. The next parts show combinations of XIL methods that combine different internally-used explainers. The fifth part presents results for combinations of XIL simultaneously utilizing penalty and reward feedback. The last part shows combinations of XIL using loss and dataset augmentation. Best results bold; higher is better

First, we report the accuracy of the vanilla model, models revised with single XIL methods, and models revised with a combination of XIL methods in Tab. 3. In terms of accuracy, applying XIL with multiple explainers provides is on par with single methods, as the accuracy among all categories remains comparable on a high level, overcoming the confounder influence. The accuracy without decoy can be seen as an upper bound, for what the architecture is able to achieve in the best case, i.e. without confounding factors.

The WR scores for single XIL methods in Tab. 4 show that the XIL methods mainly improve WR scores for the internally-used explainer. Additionally, one can observe that HINT alone does not improve the WR scores.

When combining XIL methods with different explainers (Tabs. 5 and 6), one can observe that the WR scores are low for the internally optimized explainers, but they also improve for the other explanation methods. Tab. 7 depicts that the combination of different reward strategies can be beneficial as well. Especially noteworthy is the combination of RRR-G and HINT, which utilizes the same internal explainer, but the different feedback strategies provide substantial improvements over the other individual methods. Lastly, the combination of loss-based XIL methods with dataset augmentation strategies (Tab. 8) does not show a clear tendency. While it substantially improves the explanation quality for, e.g., the combination of RRR-G and CE, it does not for the combination of RRR and CE. Here, we motivate again further research in this direction.

|  | Vanilla | RRR | RRR-G | RBR | CDEP | HINT | CE |
|---|---|---|---|---|---|---|---|
| IG | $23.1 \pm 3.8$ | $\mathbf{0.0} \pm 0.0$ | $11.9 \pm 2.1$ | $2.0 \pm 1.3$ | $15.0 \pm 1.5$ | $21.9 \pm 3.1$ | $7.3 \pm 1.4$ |
| GradCAM | $38.7 \pm 4.6$ | $13.3 \pm 2.0$ | $\mathbf{1.5} \pm 0.8$ | $15.2 \pm 3.8$ | $27.8 \pm 3.8$ | $46.8 \pm 1.1$ | $14.7 \pm 2.9$ |
| IntGrad | $41.3 \pm 2.2$ | $\mathbf{0.0} \pm 0.0$ | $21.0 \pm 3.4$ | $17.8 \pm 5.4$ | $6.8 \pm 2.7$ | $40.3 \pm 2.5$ | $19.0 \pm 1.4$ |
| LIME | $59.8 \pm 2.0$ | $\mathbf{32.1} \pm 0.4$ | $33.3 \pm 2.8$ | $37.7 \pm 3.0$ | $37.9 \pm 3.7$ | $53.8 \pm 2.0$ | $36.9 \pm 0.6$ |

(a) DecoyMNIST

|  | Vanilla | RRR | RRR-G | RBR | CDEP | HINT | CE |
|---|---|---|---|---|---|---|---|
| IG | $25.0 \pm 1.9$ | $\mathbf{0.0} \pm 0.0$ | $2.1 \pm 0.4$ | $6.0 \pm 1.4$ | $15.9 \pm 4.5$ | $29.4 \pm 3.3$ | $8.1 \pm 0.4$ |
| GradCAM | $34.8 \pm 1.4$ | $24.2 \pm 4.1$ | $\mathbf{4.6} \pm 0.9$ | $16.0 \pm 4.8$ | $39.1 \pm 1.7$ | $27.8 \pm 2.9$ | $24.4 \pm 0.9$ |
| IntGrad | $38.9 \pm 2.5$ | $\mathbf{0.0} \pm 0.0$ | $28.8 \pm 1.9$ | $19.6 \pm 5.1$ | $12.5 \pm 4.9$ | $38.3 \pm 1.0$ | $20.8 \pm 1.7$ |
| LIME | $57.6 \pm 0.8$ | $\mathbf{27.4} \pm 0.7$ | $38.1 \pm 4.5$ | $34.9 \pm 1.4$ | $40.2 \pm 6.5$ | $51.4 \pm 3.5$ | $31.1 \pm 0.6$ |

(b) DecoyFMNIST

Table 4: Mean WR scores [%] and standard deviations (5 runs) on Decoy(F)MNIST. The WR scores show that XIL mainly improves the explanation quality for internally-used explainers. The scores remain high for other explainers. Best values bold; lower is better.

|  | RRR+RRR-G | RRR+RBR | RRR-G+RBR | RRR+RRR-G+RBR |
|---|---|---|---|---|
| IG | $\mathbf{0.0} \pm 0.0$ | $\mathbf{0.0} \pm 0.0$ | $0.9 \pm 0.5$ | $\mathbf{0.0} \pm 0.0$ |
| GradCAM | $2.7 \pm 2.1$ | $21.2 \pm 2.7$ | $2.5 \pm 1.2$ | $\mathbf{2.3} \pm 0.4$ |
| IntGrad | $\mathbf{0.0} \pm 0.0$ | $\mathbf{0.0} \pm 0.0$ | $16.2 \pm 0.6$ | $\mathbf{0.0} \pm 0.0$ |
| LIME | $27.7 \pm 1.8$ | $27.1 \pm 0.5$ | $30.9 \pm 1.1$ | $\mathbf{25.6} \pm 0.8$ |

Table 5: Mean WR scores [%] and standard deviations (5 runs) on DecoyFMNIST. The columns depict combinations of explainers used for XIL. The WR scores of combined methods are lower than for methods based on single explainers (*cf.* Tab. 4b). Best values bold; lower is better.

|  | RRR+CDEP | RBR+CDEP | RRR-G+CDEP |
|---|---|---|---|
| IG | $\mathbf{0.0} \pm 0.0$ | $2.1 \pm 0.9$ | $4.9 \pm 2.1$ |
| GradCAM | $18.5 \pm 4.3$ | $19.6 \pm 5.1$ | $\mathbf{2.1} \pm 1.0$ |
| IntGrad | $0.2 \pm 0.1$ | $6.5 \pm 0.9$ | $11.0 \pm 4.4$ |
| LIME | $29.2 \pm 0.7$ | $30.8 \pm 1.8$ | $33.1 \pm 3.9$ |

(a) DecoyMNIST

|  | RRR+CDEP | RBR+CDEP | RRR-G+CDEP |
|---|---|---|---|
| IG | $0.3 \pm 0.2$ | $1.9 \pm 0.6$ | $7.5 \pm 1.3$ |
| GradCAM | $25.4 \pm 2.1$ | $35.4 \pm 2.8$ | $3.3 \pm 0.5$ |
| IntGrad | $1.1 \pm 0.2$ | $2.9 \pm 0.8$ | $6.0 \pm 3.5$ |
| LIME | $28.8 \pm 1.0$ | $26.3 \pm 0.8$ | $31.3 \pm 2.5$ |

(b) DecoyFMNIST

Table 6: Mean WR scores [%] and standard deviations (5 runs) on Decoy(F)MNIST for combinations of XIL methods with different internally-used explainers. One can observe that the scores are lower for combined methods compared to using single methods only. Best values bold; lower is better.

|  | RRR-G+HINT | CDEP+HINT | RRR+HINT | RBR+HINT |
|---|---|---|---|---|
| IG | $1.0 \pm 0.5$ | $0.3 \pm 0.1$ | $\mathbf{0.0} \pm 0.0$ | $0.8 \pm 0.4$ |
| GradCAM | $\mathbf{3.0} \pm 0.3$ | $3.2 \pm 0.2$ | $5.3 \pm 1.0$ | $3.9 \pm 1.0$ |
| IntGrad | $8.5 \pm 0.8$ | $5.6 \pm 2.8$ | $\mathbf{1.4} \pm 1.0$ | $8.5 \pm 1.4$ |
| LIME | $30.1 \pm 5.3$ | $31.0 \pm 0.5$ | $27.4 \pm 1.2$ | $\mathbf{27.3} \pm 0.9$ |

(a) DecoyMNIST

|  | RRR-G+HINT | CDEP+HINT | RRR+HINT | RBR+HINT |
|---|---|---|---|---|
| IG | $5.2 \pm 1.6$ | $15.4 \pm 4.0$ | $\mathbf{0.0} \pm 0.0$ | $1.2 \pm 0.1$ |
| GradCAM | $\mathbf{2.0} \pm 0.6$ | $12.8 \pm 1.5$ | $3.6 \pm 0.4$ | $4.2 \pm 1.1$ |
| IntGrad | $26.5 \pm 3.4$ | $14.3 \pm 5.2$ | $\mathbf{0.3} \pm 0.2$ | $16.6 \pm 2.8$ |
| LIME | $38.5 \pm 3.8$ | $38.3 \pm 4.0$ | $\mathbf{22.6} \pm 0.6$ | $30.4 \pm 0.9$ |

(b) DecoyFMNIST

Table 7: Mean WR scores [%] and standard deviations (5 runs) on Decoy(F)MNIST for combinations of XIL methods with different reinforcement strategies. Here, the penalty (RRR-G, CDEP, RRR, RBR) and reward strategy (HINT) are combined. One can observe that the scores are lower for combined methods compared to using single methods only. The first combination (RRR-G and HINT) is of special interest, as it uses the same explainer internally, but improves the non-internally used scores as well. Best values bold; lower is better.

|  | RRR+CE | RBR+CE | RRR-G+CE | CDEP+CE | HINT+CE |
|---|---|---|---|---|---|
| IG | $\mathbf{0.0} \pm 0.0$ | $3.5 \pm 1.8$ | $3.2 \pm 0.4$ | $6.6 \pm 1.5$ | $0.9 \pm 0.3$ |
| GradCAM | $11.7 \pm 1.3$ | $14.8 \pm 1.9$ | $3.2 \pm 1.0$ | $\mathbf{1.8} \pm 2.5$ | $5.8 \pm 0.7$ |
| IntGrad | $\mathbf{0.0} \pm 0.0$ | $9.1 \pm 7.5$ | $16.9 \pm 7.4$ | $10.2 \pm 1.4$ | $14.8 \pm 0.8$ |
| LIME | $29.6 \pm 1.3$ | $29.9 \pm 5.2$ | $30.2 \pm 0.5$ | $31.2 \pm 2.1$ | $\mathbf{27.3} \pm 0.5$ |

(a) DecoyMNIST

|  | RRR+CE | RBR+CE | RRR-G+CE | CDEP+CE | HINT+CE |
|---|---|---|---|---|---|
| IG | $\mathbf{0.0} \pm 0.0$ | $5.9 \pm 1.0$ | $1.0 \pm 0.3$ | $5.9 \pm 0.7$ | $0.1 \pm 0.0$ |
| GradCAM | $21.9 \pm 2.7$ | $24.8 \pm 3.8$ | $\mathbf{4.4} \pm 1.0$ | $20.2 \pm 6.9$ | $7.5 \pm 1.6$ |
| IntGrad | $\mathbf{0.1} \pm 0.0$ | $21.3 \pm 3.0$ | $27.6 \pm 1.1$ | $18.7 \pm 3.7$ | $22.9 \pm 2.5$ |
| LIME | $28.6 \pm 0.6$ | $31.7 \pm 1.2$ | $32.8 \pm 1.3$ | $32.0 \pm 1.2$ | $\mathbf{28.1} \pm 0.7$ |

(b) DecoyFMNIST

Table 8: Mean WR scores [%] and standard deviations (5 runs) on Decoy(F)MNIST for combinations of XIL methods with different revision strategies. Here, the loss (RRR, CDEP, RRR-G, RBR, HINT) and dataset augmentation strategy (CE) are combined. One can observe that the scores do not significantly improve when combining both strategies. Best values bold; lower is better.

