# OpenReview forum: "One Explanation Does Not Fit XIL"
_ICLR.cc/2023/TinyPapers — Submitted to Tiny Papers @ ICLR 2023_

### Official Review · Reviewer_zkqe · 2023-03-18

**Confidence:** 4

**Summary Of Contributions:**

The paper investigates the use of multiple explanation methods simultaneously.

**Rating:**

Clear, Correct, and Reproducible (CCR): a submission which meets the reviewing criteria

**Strengths And Weaknesses:**

The paper focuses on addressing the issue of shortcut learning in machine learning models by employing user feedback on model explanations through the XIL framework. The paper investigates the use of multiple explanation methods simultaneously for model revision through XIL. The authors identify that using only one explanation method may not be sufficient and propose considering multiple ones for better results.

Summary of Strengths:
* The experimental results presented in the paper show that using multiple explanation methods can improve the XIL framework's model revision process.

Summary of Weaknesses:
* Novelty: It seems to me that the only novel design in this framework is that the model uses multiple explanation loss, instead of a single one
* The experimental results presented in the paper are limited to a small set of explanation methods and datasets, and it is unclear whether the findings generalize to other explanation methods or datasets.


**Suggested Changes:**

The paper could benefit from providing a more detailed explanation of the XIL framework for readers who are not familiar with it.
The authors should improve the novelty of the paper
There are a few typos in the paper, such as "revise a model by employing" (should be "revising") and "\textit{one explanation does not fit XIL}" (should be "one explanation does not fit in XIL").

---

> ### Author Response · Authors · 2023-05-30
> **Revision**
>
> Thanks for your valuable feedback.
>
> We revised our manuscript by generally fixing typos and providing a broader introduction and explanation of XIL.

---

### Official Review · Reviewer_FVkG · 2023-03-31

**Confidence:** 4

**Summary Of Contributions:**

The paper is on improving the explanatory interactive machine learning (XIL) framework for deep model revision. The authors show that using only one explainability technique is not a robust approach to model revising. Doing so may not decrease the wrong reasoning (WR) rate with respect to the other XAI methods.

**Rating:**

Clear, Correct, and Reproducible (CCR): a submission which meets the reviewing criteria

**Strengths And Weaknesses:**

Strengths:
1. The paper is very well written and systematic. The ideas are nicely conveyed.
2. The paper provides extensive experimentation and show that using only one XAI method in interactive machine learning (XIL) framework may still give high WR rate with respect to other methods.
3. They show that, using combination of different XAI methods can reduce WR with respect to multiple XAI methods.

Weaknesses:
1. There is no significant novelty in the study.
2. The concept of using multiple XAI methods and trying to reduce WR rate with respect to all of them may not be a good idea as the different XAI methods are inherently different and use different principles to find the explanations (read: "Which explanation should i choose? a function approximation perspective to characterizing post hoc explanations", NeurIPS 2022). Hence, it is fine to have less error with respect to one XAI method and to have more error with respect to other. Rather, what is more important is to find the right XAI methods given a trained black box network and given a problem setting and trying to reduce WR with respect to them.
3. For example, (comapring the rows corresponding to LIME in Table 1b and Table 2) even using multiple XAI methods doesn't help decrease the WR for LIME. This is maybe because LIME (a perturbation-based technique) is inherently different than IG, IntGrad and GradCAM (different gradient-based methods). However, it is appriciable that the WR rates are uniformly reduced for all the IG, IntGrad and GradCAM (different gradient-based methods) when the combination of multiple XIL frameworks are used.

**Suggested Changes:**

1. Can you use XIL frameworks that use perturbation XAI methods (LIME, RISE(Petsiuk et al 2018) etc.) rather than gradient based methods like IG, IntGrad and GradCAM? It will be interesting to see if combining such multiple XIL methods help to uniformly reduce WR with respect to multiple perturbation based methods.

---

> ### Author Response · Authors · 2023-05-30
> **Revision**
>
> Thanks for your valuable feedback.
>
> The general setup of XIL is not limited to gradient-based XAI methods. So, (e.g.) perturbation-based methods could be incorporated into this loop as well. We like this idea very much and consider it as future work.

---

### Author Response · Authors · 2023-05-15
**Revision**

Dear reviewers,

thanks for the acceptance to the ICLR 23 tiny paper track and for your reviews including suggestions for improvement.  We updated our manuscript accordingly and hope it suits you well. Please feel free to reach out at any time if you have further suggestions.

---

### Author Response · Authors · 2023-05-30
**Archival**

We are happy to opt-in for archival.

---

### Meta-Review · Area_Chair_djbA · 2023-04-04

**Recommendation:** Invite to archive
**Confidence:** 4

**Metareview:**

The paper investigates the effectiveness of using multiple explanation methods in the explanatory interactive machine learning (XIL) framework for deep model revision. Reviewers agree that the paper is clear, correct, and reproducible (CCR) and present strengths in presenting experimental results that show the effectiveness of using multiple explanation methods. The weaknesses are the lack of novelty and potential issues with combining inherently different XAI methods.

**Summary:**

The paper investigates the effectiveness of using multiple explanation methods in the explanatory interactive machine learning (XIL) framework for deep model revision.

**Comments And Feedback To The Authors:**

It is advisable for the authors to
1. Improve the novelty of the paper
2. Add a more detailed explanation of the XIL framework for readers who are not familiar with it
3. Correct any typos in the paper.

**Reason For Not Giving A Higher Recommendation:**

The paper presents relatively sufficient experimental results that demonstrate the effectiveness of using multiple explanation methods.

**Reason For Not Giving A Lower Recommendation:**

Both reviewers agree that the paper is clear, correct, and reproducible (CCR) and present strengths in presenting experimental results that show the effectiveness of using multiple explanation methods.

---

### Decision · Program_Chairs · 2023-04-08

Invite to archive